# Event-based, Recursive Neural Networks for the Extraction and Aggregation of International Alliance Relations

## Abstract

In this article, we explore how text mining can help producing reliable, high-level numerical data for multi-document relation extraction applications. We combine neural network techniques, information aggregation and visualization methods to extract event-based relations with a good tolerance to noise. We describe a search engine based on these methods, identifying the evolution of alliance and opposition relations between countries.

## 1 Introduction

Information and communication technologies have provided tools and methods to make the production of information more democratic. As a result, a vast amount of content is available, which arguably creates more noise than knowledge at the end of the day. Without hierarchical organization and contextualization, users may lack perspective to understand and assimilate the multiplicity of events that they come accross every day, and to link them to related events in the past.

In this context, we need tools for extracting, aggregating and visualizing knowledge in order to facilitate information analysis and access a variety of points of view. In this article, we take the example of the geopolitical alliances and their evolution; we extract relations from the texts and aggregate them in order to turn a set of very focused pieces of information into a broader knowledge and a bigger picture of a given situation.

The main contributions of this paper are:

- Relation extraction: we present an event-based, recursive neural network approach for identifying alliance and opposition relations between countries, at sentence level, in English. We show that adapting the models to the eventive nature of extracted relations helps existing recursive neural models. We use a precision-oriented cost function for a better later aggregation.

- Relation aggregation: we aggregate the sentence-level relations in a multi-document environment, in order to obtain a picture of the geopolitical situation and evolution on a specific subject defined by a user query (e.g., situation in Syria, nuclear proliferation, North Pole ownership). We cope with the inevitable amount of noise to produce reliable numerical data from texts.

## 2 System Overview and Dataset

### 2.1 Relation Extraction

We extract opposition (*NEG*), alliance (*POS*) or neutral (*NEU*) relations between two countries, explicitly expressed in news titles. Such relations are illustrated by the following examples:

(1) **Japan** and **China** agree to reduce tensions over Senkaku islands.  → *POS(Japan, China)*

(2) **Obama**, **Merkel** warn **Russia** against intervention.  → *NEG(U.S.A., Russia)*, *NEG(Germany, Russia)*, *POS(U.S.A., Germany)*

(3) **Serbia** prepares hero's welcome for **Putin**.  → *POS(Serbia, Russia)*

(4) **Russia** backs **Ukraine** rebel vote.  → *NEG(Russia, Ukraine)*

(5) **British** teens detained en route to **Syria**, police say.  → *NEU(Great Britain, Syria)*

Countries (arguments) can be designated by their actual names, the name of their capital or a representative person. Unlike for many knowledge extraction approaches, these relations are very changing, making the problem more difficult.

## 2.2 Relation Aggregation

We then index these relations together with the text and the date of the corresponding articles. At query time, we retrieve all titles relevant to the query, extract the candidate entities and their relations. For each day, *POS* and *NEG* relations are aggregated and a tendency of this day is obtained, as well as a trend over time. A graphical visualization of this trend is then proposed to the user.

We chose to use only titles because we can consider that a title describes only one event, and that this event can be time-stamped by the document creation time, which is not true in article content. However, this comes with a drawback : titles in English often follow some different syntactic rules than traditional sentences, as in examples 2 (coordination expressed with only a comma) and 5 (contracted reported speech). Most parsers are not trained on title sentences and will then experience a higher error rate.

## 2.3 Dataset

Country, nationality, capital name and main leader lists were extracted from the linked data repository of the CIA World Factbook, and linguistic variations of these names were collected from DBPedia (Lehmann et al., 2013). Major unions of countries (United Nations, European Union, ASEAN, African Union...) and important groups (ISIS...) are also considered. All these elements were considered as entities potentially parts of an alliance/opposition relation.

We used a corpus of 3.3 millions of English news titles from about 20 different web news sites, between 2013 and early 2016 (3,000 documents/day in average). From this corpus, we randomly selected titles containing at least two entities. Each pair of entities is an instance of the classifier, which means that one sentence can lead to several instances. Furthermore, we added the heads of the nominal phrase containing these entities, as well as "person triggers" (*man*, *minister*, etc.) in order to refine the training. For example:

(6) **EU**, **U.N.** condemn **Ukraine rebels**' election plans. → *NEG(EU, rebels)*, *NEG(UN, rebels)*, *POS(EU, Ukraine)*, *POS(UN, Ukraine)*, *POS(EU,UN)*

the word *rebels* is added to *EU*, *U.N.* and *Ukraine*, leading to five possible pairs (entities

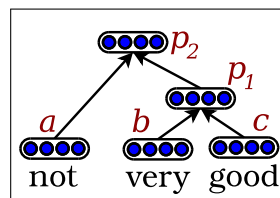

Figure 1: Recursive neural network.

from the same chunks are not considered).[1] These extra entities are expected to help the model use all possible information.

Among all these automatically collected entities, we manually annotated 5172 pairs with polarities *NEU* (2837 instances), *NEG* (1473 instances) and *POS* (862 instances), coming from 2166 different titles (between 1 and 15 pairs in a title). Only explicit relations were annotated *POS* or *NEG*, and no geopolitical knowledge was used.

## 3 Relation Extraction at Sentence Level with Recursive Neural Models

We describe now the models used for classifying relations at sentence level, as described in Sec. 2.1.

### 3.1 Recursive Neural Networks

Recursive neural networks (Socher et al., 2011, 2013) have proved useful for tasks involving long-distance relations, such as semantic relation extraction (Hashimoto et al., 2013; Li et al., 2015).

As illustrated by Figure 1 (from Socher et al. (2013)), an n-gram is parsed into a binary tree and each word is represented as a $K$-dimension vector. The representation of each parent node is computed based on its immediate children, in a bottom-up fashion, using the following equations:

$$p_1 = f(W_\mathrm{L} \cdot b + W_\mathrm{R} \cdot c); p_2 = f(W_\mathrm{L} \cdot a + W_\mathrm{R} \cdot p_1)$$

where $f(\cdot)$ is the function $tanh$, and $W_\mathrm{L}$ and $W_\mathrm{R}$ are $K \times K$ matrices. Vector $p_1$ has the same dimension as word vectors and can then be used for computing its own parent node.

The root representation will then be used as features to classify each phrase with a softmax layer.

### 3.2 Our Models

Our full-tree model (FT) is based on this exact principle.[2] For example, in Figure 2.a, the

---

[1]Note that we consider the mention of the country as representing its government position.

[2]Note that we do not report results with the tensor version of recursive NNs since they systematically decreased results.

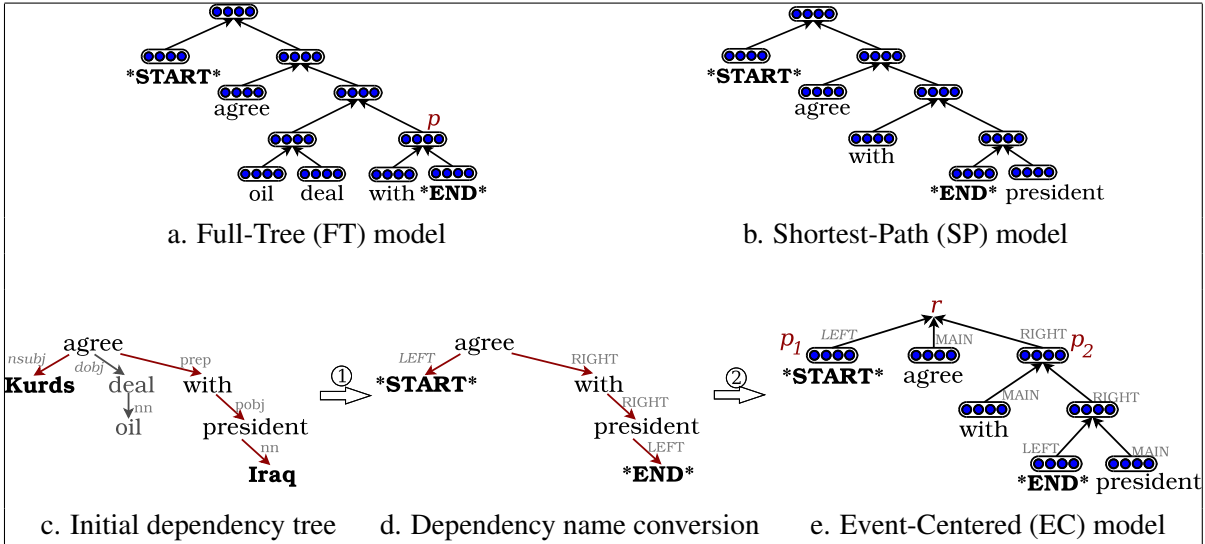

Figure 2: Different representations of the relation "**Kurds** agree oil deal with **Iraq** president".

Figure 3: Other examples of conversion from dependency graph to EC trees.

parse tree obtained with Stanford parser (Klein and Manning, 2003) is binarized, and the vector for parent $p$ is calculated from $K$-dimension word embeddings $w_{with}$ and $w_{END}$:

$$p = f(W_{\text{LEFT}} \cdot w_{with} + W_{\text{RIGHT}} \cdot w_{END} + b)$$

where $b$ is the $1 \times K$ bias vector. In a Shortest-Path (SP) model, we use the same representation but we keep only nodes and branches that are part of the shortest dependency path (see Figure 2.c then 2.b, where the "*oil deal*" is pruned but "*president*" is added), as experimented by Xu et al. (2015) for LSTM networks.

Dependency tree recursive neural networks (Socher et al., 2014) generalizes to non-binary trees by considering as many composition matrices as necessary (until one matrix per dependency type). This will allow us to manipulate the tree as desired in our Event-Centered (EC) model described in the following section.

As we do not seek for general knowledge but for changing relationships, the entities themselves are replaced by generic *START* and *END* words, as well as proper nouns that are replaced by *PN*. This guarantees that the model will really learn the relational structure rather than the specific connection between two countries. This also gives us a way to represent the entities in the structure, without having to stick to the exact path between them (as in Socher et al. (2012)) or to add positional features (as in Zeng et al. (2014)).

### 3.3 Event-Centered Modeling

A known issue of recursive models is that their deep tree structure leads to a bias towards the topmost nodes (Kalchbrenner et al., 2014). We turn this problem into a desirable feature by pushing the main event governing the relation to the top of the structure.[3] This section details this operation, illustrated by Figures 2.c, 2.d and 2.e.

At step ①, dependencies from Stanford parser are renamed by the position of the dependent w.r.t the governor (LEFT is the dependent if before the governor in the sentence, RIGHT otherwise). Only the path between entities is kept. Then (step ②), each branch node (*agree*, *with* and *president*) is extracted and converted into a leaf node, with an edge labeled MAIN going from this leaf node to the former position of the word. This label represents the fact that the word is the governor of its tree sibling in a dependency.

---
[3]Therefore, we do not use any node weighting nor normalization by the number of words underneath them, unlike in previous work (Socher et al., 2011, 2014).

We obtain a structure that we call Event-Centered (EC), that can be processed by a dependency tree recursive model. For example, in Figure 2.e, the root vector $r$ is calculated as follows:

$$r = f(W_{\text{LEFT}} \cdot p_1 + W_{\text{MAIN}} \cdot wagree + W_{\text{RIGHT}} \cdot p_2 + b)$$

The top node of EC trees can have three children, all the others have only two. Only three matrices ($W_{\text{MAIN}}, W_{\text{LEFT}}, W_{\text{RIGHT}}$) are created, keeping the number of parameters reasonable. Figure 3 illustrates two more examples of conversion.

As our relations are triggered by events, we expect at least one event word to be part of the dependency path between the two entities (we call an *event word* either a non-state verb or a noun tagged as *act* or *event* in WordNet (Miller, 1995)). It may not be the case, either because of a specific structure of the sentence or because of a parser error, mainly due to the speficities of titles or to traditional bad prepositional attachments. We now describe how we force the path to contain an event word.

If the dependency path from $E_1$ to $E_2$ does not contain any event word, we extend this path by collecting common ancestors until we meet one. For example, in the following sentence:

(7) **Japan** and **China** agree to reduce tensions over Senkaku islands

the path is turned from *Japan–conj–China* to *agree—Japan—China*.[4]

If no event word is found, then the closest event word to $E_1$ and $E_2$ (or as a last resort, the root node) is attached to the entities. This is the case in

(8) **ISIS** militants attack **Syrian** town near Turkey border

with the (erroneous) initial dependency graph

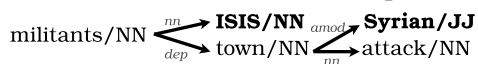

where *attack* is not in the path, but is added because *attack* is an event word. The graph is then centered on this event so that we obtain a tree as described before:

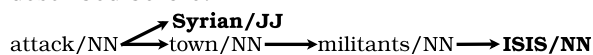

Using positional relations instead of real dependency names allows to twist the graph as desired to save what can be saved from an erroneous graph.

---

[4]Note that we used the uncollapsed version of Stanford dependency graphs in order to keep word vectors for prepositions. Not doing so decreases dramatically the results.

### 3.4 Precision-Oriented Cost Function

Our final application aggregates the sentence-based relations and uses a large corpus where the same relation on the same day will be expressed many times in different manners. This is why, faced with the traditional trade-off dilemma between precision and recall, we want to favor precision, for two reasons:

- Aggregating results with poor precision tends to lead to wrong aggregated values.

- The corpus redundancy will help to gather useful information even with a classifier with lower recall.

We then bias the classical cross-entropy error function by adding a penalty to false positives (i.e. *POS* or *NEG* erroneous predictions). For each relation $r$ represented by the parameter vector $\theta$, the error function is then

$$E(r; \theta) = -\lambda_{FP} \left( \sum_{k=1}^{3} t_k \log d_k(r; \theta) \right)$$

where $k$ represents each of the three classes *POS*, *NEG* and *NEU*, $t_k$ is the $k^{th}$ element of the multinomial target label distribution $t$ for a relation, $d_k(r; \theta)$ the 3-dimensional output of the softmax layer. Finally, $\lambda_{FP} = 1$ if the prediction is correct or false negative (*NEU* instead of *POS* or *NEG*) and $\lambda_{FP} = \alpha$ (with $\alpha > 1$) if the prediction is a false positive, so that it is more costly. This can also be seen as an extra regularization parameter on the sum of all false positive predictions. In our experiments, we choose $\alpha$ such that it guarantees a precision higher than 80% on dev *POS* and *NEG* classes and maximizes the overall accuracy.

### 3.5 Extra Features

In addition to the top vector of the recursive model, we add the following features to the softmax classifier: average of the three words before $E_1$ (resp. after $E_2$); concatenation of the three words after $E_1$ (resp. before $E_2$); average of all event words (as defined above) between $E_1$ and $E_2$, as well as in the sentence. This leads to a total of $10 \times K$ extra parameters.

## 4 Classification Experiments and Results

### 4.1 Baselines

We evaluate this sentence-level extraction step and we compare our different models with:

**SVM.** A SVM two-step model using a sentiment lexicon and hand-crafted resources for alliance/opposition triggers, together with traditional positional, lexical and syntactic features. The first step classifies between neutral and non-neutral relations, while the second step distinguishes between positive and negative relations.

**LSTM.** A Long Short Term Memory (LSTM) model (Hochreiter and Schmidhuber, 1997) with softmax classifier.

**TABARI.** A rule engine (Schrodt and Yonamine, 2012) and a set of rules oriented toward inter-state behavior (Schrodt et al., 2008), composed of 284 relation labels that we mapped to either positive, negative and neutral relations. As the rules date back to 2008, we enriched the actor list by the entities from our corpus.

### 4.2 Implementation

We report results for two different configurations:

- Models FT, SP and EC with a softmax layer on the top node of the relation representation. The parameters are then the word embeddings ($V \times K$ parameters), the two or three $K \times K$ composition matrices, the softmax matrix ($K \times 3$) and bias vectors ($K$ each).

- The same models with the extra features described in Section 3.5 ($10 \times K$ parameters). These are called respectively FT+F, SP+F and EC+F, with or without precision-oriented cost function.

We pre-train word vectors with word2vec (Mikolov et al., 2013) applied to our entire article corpus. For all models, we use a development set for early stopping and hyper-parameter tuning (regularization weights, word vector size – set to $K = 50$ – and parameter $\alpha$). We failed at finding an $\alpha$ meeting the requirements for the FT model. All these models are implemented with Theano (Bastien et al., 2012) and L-BFGS-B optimization, and tested on a test set of 453 relations.

### 4.3 Classification Results

Table 1 reports the results obtained by the described classifiers. Precision and recall values are

| System | Prec. | Rec. | Acc. |
|---|---|---|---|
| *Baselines* | | | |
| TABARI | 0.573 | 0.564 | 0.569 |
| SVM | 0.599 | 0.673 | 0.606 |
| LSTM | 0.645 | 0.573 | 0.663 |
| *Top node only* | | | |
| FT | 0.719 | 0.613 | 0.703 |
| SP | 0.726 | 0.600 | 0.709 |
| EC | 0.698 | 0.587 | 0.703 |
| *Unbiased cost function ($\alpha = 1$)* | | | |
| FT+F | 0.749 | 0.636 | 0.733 |
| SP+F | 0.787 | 0.609 | 0.742 |
| EC+F | 0.813 | **0.676** | **0.771** |
| *Precision-oriented cost function ($\alpha > 1$)* | | | |
| SP+F$_{\alpha=2.6}$ | 0.809 | 0.564 | 0.734 |
| EC+F$_{\alpha=2.3}$ | **0.827** | 0.636 | 0.760 |

Table 1: Sentence-level classifier results.

based on *NEG* and *POS* classes[5], accuracy is on the three classes.

SP and EC models do not improve the results when only the top node of the structure is used as a feature for the softmax classifier. This can be explained by the fact that more sentence words are not represented at all in SP and EC structure, while FT contains all words between entities. However, with the extra content features, the EC accuracy is much higher, showing that a more focused tree structure can make the difference when the vocabulary gap is filled by representing missing words with simple average or concatenation.

Even the unbiased cost function leads to a precision much higher than recall. Our precision-oriented cost function with a tuned $\alpha$ value leads to a small precision gain with a small recall loss.

The system TABARI does not perform well on this dataset. This can be due to the fact that the dataset addresses a very large variety of subjects and geographic areas; TABARI gets better performances on specific topics with dedicated rules.

In what follows, we use the model EC+F$_{\alpha=2.3}$.

## 5 Time-Aware Aggregation and Visualization

We describe here how we aggregate the sentence-level relations. A Solr search engine (Kuć, 2013)

---

[5]*NEU* is much more represented and useless for aggregation; integrating it to prec. and rec. values would increase them artificially, making a *NEU-only* classifier the best one.

is built from time-stamped alliance relations. Each sentence containing at least one relation is indexed with its stemmed content, the countries involved in relations, the relations themselves and the document creation date. A typical query is then composed of a few keywords representing the topic, a temporal interval (minimum of maximum dates) and zero or more country names on which the user wants to restrict relation extraction.

### 5.1 Aggregating Information

All sentences returned by Solr, without any number limit, are aggregated. For all pairs of considered countries, inside the same day $d$, if the numbers of *POS* and *NEG* relations between the two countries are respectively $P(d)$ and $N(d)$, then the weight for the pair and the day $d$ is:

$$w(d) = \log\left(\frac{1 + P(d)}{1 + N(d)}\right)$$

which leads to a number between $-\infty$ and $+\infty$, where $w = 0$ means that the relation is neutral, $w < 0$ that the relation is an opposition, and $w > 0$ that the relation is an alliance.

The noise is then reduced by a weighted mean smoothing over a temporal window of $W$ days:

$$sw(d) = \frac{\sum_{j=-n}^{n} \frac{w(d+j)}{abs(j)+1}}{\sum_{j=-n}^{n} \frac{1}{abs(j)+1}}$$

where n is a half-window ($n = floor(\frac{W}{2})$)

For example, for a 5-day window:

$$sw(d) = \frac{\frac{w(d-2)}{3} + \frac{w(d-1)}{2} + w(d) + \frac{w(d+1)}{2} + \frac{w(d+2)}{3}}{\frac{1}{3} + \frac{1}{2} + 1 + \frac{1}{2} + \frac{1}{3}}$$

### 5.2 Visualization

#### 5.2.1 Time-Series Plot

For bilateral relations (field *countries* containing two items), we provide the users with a time-series plot representing $sw(d)$, and show them on demand the sentences which led to this value. Figure 4 shows an example of results concerning the relations between United States and Russia ("*countries:United States AND Russia*") concerning the situation in Syria ("*keywords:syria*").

#### 5.2.2 Force-directed Graph

When zero or one country is specified by the user, we generate a graph of countries, where the distance between vertices (*i.e.*, countries) reflects the opposition between the countries in a given time

| Relation | $sw(d)$ | $sw'(d)$ | Relation | $sw(d)$ | $sw'(d)$ |
|----------|---------|----------|----------|---------|----------|
| $A_1$ - $A_2$ | 1 | 0.27 | $B$ - $A_1$ | -1 | 0.73 |
| $A_1$ - $A_3$ | 2 | 0.12 | $B$ - $A_2$ | -2.3 | 0.91 |
| $A_2$ - $A_3$ | 3.5 | 0.03 | $B$ - $A_3$ | -3 | 0.95 |

Table 2: Examples of logarithmic weights $sw(d)$ and their logistic adjusments $sw'(d)$.

span (generally, larger that a single day). The problem here is that our classifier provides a relation between two countries independently of the others; it is therefore very unlikely that these values can be used as distance values in the whole graph (especially, triangle inequality will not be respected). For this reason, we use the Barnes-Hut force-directed layout algorithm (Barnes and Hut, 1986), where a value between two vertices is considered as a repulsive force. The algorithm tries to minimize the energy spent to keep the whole graph together, and then to adapt the distances between vertices so that the rendering on a 2D plan is possible.

For this algorithm, we need to transform our weights $sw(d)$ into positive numbers (repulsive forces). We also need to damp the noise and the variations of the weights that are introduced by the volume of data. This problem is illustrated by Table 2, where three countries $A_1$, $A_2$ and $A_3$ are allies against another one $B$ on a given topic and a given time span. The weights $sw(d)$ of the three relations between allies are positive but vary between 1 and 3.5. As a result, $A_1$ is finally closer to $B$ (difference of 2) than to $A_2$ (difference of 2.5), which is not desirable.

This kind of effects can be corrected by a "S-shaped", logistic function, that models a level of saturation after an approximately exponential growth (or, in our case, decrease):

$$sw'(d) = 1 - \frac{1}{1 + e^{-sw(d)}} \qquad (9)$$

This function levels off high weights (both negative and positive), increases differences between positive and negative values and thus helps reducing noise without having to discretize values arbitrarily. Resulting numbers are all positive, between 0 (strong alliance, attraction in the graph) and 1 (strong opposition, repulsion in the graph).

Figure 5 shows an example of such graphs obtained by the force-directed algorithm, with the keyword *syria*.

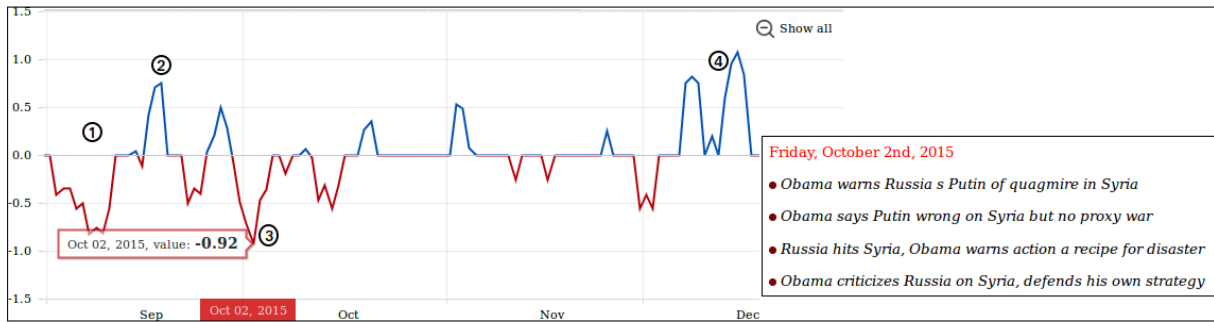

Figure 4: Example of plot produced by the system for bilateral relations between United States and Russia on the query "Syria", on a daily scale. The right frame shows sentences corresponding to the user-selected date (Oct. 2, 2015). Circled numbers have been manually added to the screenshot to make some peaks clearer. ① Russian military involvement grows in Syria (bad relation, $sw(d) < 0$); ② US and Russian defence chiefs start discussing (better relation, $sw(d) > 0$); ③ Russian airstrikes in Syria; ④ Optimism after U.S. Secretary of State's visit to Moscow.

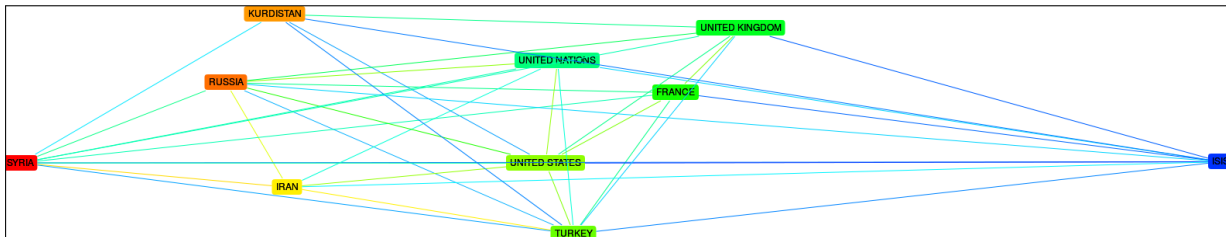

Figure 5: Example of graph produced by the system for relations between different states on the query "syria", for the year 2015. Edge colors indicate the kind of relation (from dark red for strong alliance to dark blue for strong opposition), and vertice colors reflects proximity of countries with each other.

## 5.3 Evaluation

Evaluating the relevance of trends suggested by a plot or a dynamic map is very subjective and would require a high level of expertise in each domain concerned by the tested queries. That is why we opted for a protocol that is at the same time more objective and easier to conduct:

1. We chose 14 queries with the following information: names of two countries (or unions of countries) involved in the relation, and an optional keyword-based thematic restriction. We selected queries having potentially a high density of extracted relations (*e.g.* North Korea *vs.* South Korea, or Russia *vs.* United Nations on "Syria"), as well as sparser topics (China *vs.* Japan on maritime affairs).

2. On the resulting plot, we selected up to 5 strong peaks — $abs(sw(d)) > 1$ — and 5 weak peaks — $abs(sw(d)) \leq 1$. This led to a total of 97 relations to evaluate.

3. For each of these peaks, we estimated whether the polarity of the peak was relevant or not, from a different news article collection. This is still a heavy task, which explains the low number of tested instances.

The accuracy of strong peaks is 0.93, making them highly reliable. Accuracy of weak peaks is 0.752.

## 6 Related Work

**TDT and Opinion Mining.** The idea of automatically finding topically related material in streams of newswire data goes back to Topic Detection and Tracking evaluation campaigns (Allan, 2002). These campaigns and the follow-up research aim at developing algorithms for identifying and organizing events described in documents.

Our work contributes to this global research effort and explores the quantitative aspects of knowledge that can be extracted from textual documents. On this point, as well as on the topic

of alliance and opposition relations, this can be connected to sentiment analysis and opinion mining (Liu, 2012). Different opinions must be aggregated, not only for getting more information, as in multidocument information extraction, but also for balancing positive and negative statements and conclude about the average opinion.

Fewer research works focused on extracting opinion relations between entities, including Gottipati et al. (2013); Yang and Cardie (2013); Johansson and Moschitti (2013), that can be related to our work.

**International Relations.** TABARI (a.k.a. KEDS) is an open-source program extracting events concerning international relations from news texts (Schrodt and Yonamine, 2012). This rule-based system makes extensive use of hand-crafted dictionaries and patterns. It has been widely used in the field of political science, for example for modeling dependencies in international relations networks (Hoff and Ward, 2004), as well as in the GDELT Project.[6] Several versions have been proposed for studying different aspects of international relations (Levant, Turkey, Balkans, Political Instability Task Force, etc.) and a more general-purpose set of rules, CAMEO, has also been developed (Schrodt et al., 2008). We compare it to our approach, after a few necessary adaptations (Section 4.1). More recently, (Boschee et al., 2013) presented a proprietary system and compare to TABARI as well.

With very different approaches, O'Connor et al. (2013) describe an unsupervised model for extracting events between political actors, and Gerrish (2013) presents a time-series topic model of foreign affairs. Similar aspects with our work are the detection of conflicts, as well as the interest in the temporal evolution of these conflicts and a general approach for studying specific situations. However, the purpose and output (distributions of topical terms) are different.

Finally, Chambers et al. (2015) describe an identification of political sentiment between nations at the citizen level, by using a large Twitter dataset. Their work is more directly related to traditional sentiment analysis and does not rely specifically on geopolitical events, even if the observed trends can follow these events.

---

[6] http://gdeltproject.org

**Neural Networks for Relation Extraction.** Several neural network models have been used for sentiment analysis and relation extraction. Examples are convolutional networks (Zeng et al., 2014; Nogueira dos Santos et al., 2015), recurrent networks such as LSTM (Xu et al., 2015) or recursive networks (Socher et al., 2011, 2012, 2013). Li et al. (2015) shows that using tree structures, as do recursive models, helps in tasks where long-distance relations are involved, such as semantic relation extraction. The results described in this paper confirm this conclusion.

Our precision-oriented approach relies on linguistic variation and redundancy in a large amount of documents to ensure a good coverage (Yates, 2007). It is related to some works in web question-answering (Dumais et al., 2002; M.Banko et al., 2002; Magnini et al., 2002), temporal information aggregation (Kessler et al., 2012) or opinion mining (Turney, 2002; Hu and Liu, 2004).

## 7 Conclusion & Perspectives

The advances in information extraction techniques and the increase of computing power make possible quite deep analysis of very large collections of texts. In this paper, we propose to take advantage of this progress to aggregate a large volume of extracted relations and to turn this knowledge into numerical data. We also propose a tool identifying the evolution of alliance and opposition relations between countries, on a specific subject.

Most information conveyed every day remains textual and unstructured. With such applications, a wide new range of knowledge can become available, in many different fields, which would bring high added value to the final users, in terms of aggregation, contextualization and hierarchical organization of information. We see our international relations example as a use case study of this effort.

We now intend to focus on reducing the necessary amount of supervision by using distant supervision, so that different subjects could be treated with less effort. Also, such study on country relationships should be multilingual in order to consider the cultural dimension as well.

This article is accompanied with supplementary material: the annotated corpus, the FT, SP and EC versions of all relations (JSON format), the script converting text into each of these versions, and the source for our recursive models.

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
