# Peer review of "Event-based, Recursive Neural Networks for the Extraction and Aggregation of International Alliance Relations"

_ACL 2017 — decision unknown_

[Official Review · Reviewer 1 · rating 2 · confidence 4]
soundness 5 · originality 5 · clarity 2 · impact 3 · substance 4 · appropriateness 5 · meaningful comparison 3 · presentation format Poster

- Strengths: Useful modeling contribution, and potentially useful annotated
data, for an important problem -- event extraction for the relationships
between countries as expressed in news text.

- Weaknesses: Many points are not explained well in the paper. 

- General Discussion:

This work tackles an important and interesting event extraction problem --
identifying positive and negative interactions between pairs of countries in
the world (or rather, between actors affiliated with countries).  The primary
contribution is an application of supervised, structured neural network models
for sentence-level event/relation extraction.  While previous work has examined
tasks in the overall area, to my knowledge there has not been any publicly
availble sentence-level annotated data for the problem -- the authors here make
a contribution as well by annotating some data included with the submission; if
it is released, it could be useful for future researchers in this area.

The proposed models -- which seem to be an application of various
tree-structured recursive neural network models -- demonstrate a nice
performance increase compared to a fairly convincing, broad set of baselines
(if we are able to trust them; see below).  The paper also presents a manual
evaluation of the inferred time series from a news corpus which is nice to see.

I'm torn about this paper.  The problem is a terrific one and the application
of the recursive models seems like a contribution to this problem. 
Unfortunately, many aspects of the models, experimentation, and evaluation are
not explained very well.  The same work, with a more carefully written paper,
could be really great.

Some notes:

- Baselines need more explanation.  For example, the sentiment lexicon is not
explained for the SVM.                    The LSTM classifier is left highly
unspecified
(L407-409) -- there are multiple different architectures to use an LSTM for
classification.  How was it trained?  Is there a reference for the approach? 
Are the authors using off-the-shelf code (in which case, please refer and cite,
which would also make it easier for the reader to understand and replicate if
necessary)?  It would be impossible to replicate based on the two-line
explanation here.  

- (The supplied code does not seem to include the baselines, just the recursive
NN models.  It's great the authors supplied code for part of the system so I
don't want to penalize them for missing it -- but this is relevant since the
paper itself has so few details on the baselines that they could not really be
replicated based on the explanation in the paper.)

- How were the recursive NN models trained?

- The visualization section is only a minor contribution; there isn't really
any innovation or findings about what works or doesn't work here.

Line by line:

L97-99: Unclear. Why is this problem difficult?  Compared to what? (also the
sentence is somewhat ungrammatical...)

L231 - the trees are binarized, but how?

Footnote 2 -- "the tensor version" - needs citation to explain what's being
referred to.

L314: How are non-state verbs defined?                    Does the definition of
"event
word"s
here come from any particular previous work that motivates it?                   
Please
refer to
something appropriate or related.

Footnote 4: of course the collapsed form doesn't work, because the authors
aren't using dependency labels -- the point of stanford collapsed form is to
remove prepositions from the dependeny path and instead incorporate them into
the labels.

L414: How are the CAMEO/TABARI categories mapped to positive and negative
entries?  Is performance sensitive to this mapping?  It seems like a hard task
(there are hundreds of those CAMEO categories....) Did the authors consider
using the Goldstein scaling, which has been used in political science, as well
as the cited work by O'Connor et al.?  Or is it bad to use for some reason?

L400-401: what is the sentiment lexicon and why is it appropriate for the task?

L439-440: Not clear.  "We failed at finding an alpha meeting the requirements
for the FT model."  What does that mean? What are the requirements? What did
the authors do in their attempt to find it?

L447,L470: "precision and recall values are based on NEG and POS classes". 
What does this mean?  So there's a 3x3 contingency table of gold and predicted
(POS, NEU, NEG) classes, but this sentence leaves ambiguous how precision and
recall are calculated from this information.

5.1 aggregations: this seems fine though fairly ad-hoc.  Is this temporal
smoothing function a standard one?  There's not much justification for it,
especially given something simpler like a fixed window average could have been
used.

5.2 visualizations: this seems pretty ad-hoc without much justification for the
choices.  The graph visualization shown does not seem to illustrate much. 
Should also discuss related work in 2d spatial visualization of country-country
relationships by Peter Hoff and Michael Ward.

5.3
L638-639: "unions of countries" isn't a well defined concept.  mMybe the
authors mean "international organizations"?

L646-648: how were these 5 strong and 5 weak peaks selected?  In particular,
how were they chosen if there were more than 5 such peaks?

L680-683: This needs more examples or explanation of what it means to judge the
polarity of a peak.  What does it look like if the algorithm is wrong?               
   
How
hard was this to assess?  What was agreement rate if that can be judged?

L738-740: The authors claim Gerrish and O'Connor et al. have a different
"purpose and outputs" than the authors' work.  That's not right.  Both those
works try to do both (1) extract time series or other statistical information
about the polarity of the relationships between countries, and *also* (2)
extract topical keywords to explain aspects of the relationships.  The paper
here is only concerned with #1 and less concerned with #2, but certainly the
previous work addresses #1.  It's fine to not address #2 but this last sentence
seems like a pretty odd statement.

That raises the question -- Gerrish and O'Connor both conduct evaluations with
an external database of country relations developed in political science
("MID", military interstate disputes).              Why don't the authors of this
work do
this evaluation as well?  There are various weaknesses of the MID data, but the
evaluation approach needs to be discussed or justified.